# Systemic Antibiotics and Obesity: Analyses from a Population-Based Cohort

**DOI:** 10.3390/jcm10122601

**Published:** 2021-06-12

**Authors:** So Young Park, Morena Ustulin, SangHyun Park, Kyung-Do Han, Joo Young Kim, Dong Wook Shin, Sang Youl Rhee

**Affiliations:** 1Department of Endocrinology and Metabolism, Kyung Hee University Hospital, Seoul 03080, Korea; malcoy@hanmail.net; 2Medical Science Research Institute, Kyung Hee University Medical Center, Seoul 03080, Korea; more.u88@gmail.com; 3Department of Medical Statistics, College of Medicine, Catholic University of Korea, Seoul 03080, Korea; ujk8774@naver.com; 4Department of Statistics and Actuarial Science, Soongsil University, Seoul 03080, Korea; hkd@ssu.ac.kr; 5Division of Endocrinology and Metabolism, Department of Internal Medicine, Hanil General Hospital, Seoul 03080, Korea; ppobong@daum.net; 6Supportive Care Center, Department of Family Medicine, Samsung Medical Center, Sungkyunkwan University School of Medicine, Seoul 03080, Korea; dwshin.md@gmail.com; 7Department of Endocrinology and Metabolism, Kyung Hee University School of Medicine, Seoul 03080, Korea

**Keywords:** anti-bacterial agents, obesity, cohort studies, republic of Korea

## Abstract

Background: In this study, we analyzed the association between antibiotic use and obesity and metabolic syndrome (MS) in a Korean adult population. Methods: Subjects using the Korean National Health Insurance Service sample cohort were retrospectively analyzed in 2015. The differences in obesity and metabolic syndrome (MS) status were compared and analyzed according to duration of systemic antibiotic treatment in the previous 10 years (non-users, 1st, 2nd, and 3rd tertile). Results: Subjects who used systemic antibiotics for longer periods were older, satisfied more criteria for MS, and had more comorbidities than non-users (non-users vs. 3rd tertile, *p* < 0.0001 for all). After adjusting for confounding factors, the risk of obesity was higher in subjects who used systemic antibiotics for longer periods than in non-users (non-users vs. 3rd tertile, OR (odds ratio) (95% CI (confidence interval)); 1.20 (1.12–1.38)). The criteria for MS were more satisfied in the 3rd tertile than in non-users. A higher obesity risk was also found in subjects treated with antibiotics targeting Gram-negative organisms than in those targeting Gram-positive organisms. Conclusion: The risk of obesity was higher in subjects who took systemic antibiotics more frequently. The risk was more prominent when they took antibiotics targeting Gram-negative bacteria.

## 1. Introduction

The prevalence of obesity is increasing worldwide in both adults and children [1,2]. Obesity is associated with several health problems such as hypertension (HTN), diabetes mellitus (DM), dyslipidemia, sleep apnea, and osteoarthritis. Obesity is related to lifestyle factors such as eating habits, physical activity, and genetic factors. In addition, there is evidence that alterations in the gut microbiome can affect some physiological processes that influence obesity status [3,4,5].

An imbalance in the gut microbiome decreases the synthesis of angiopoietin-like protein 4 (ANGPTL4), which inhibits lipoprotein lipase and causes an increase in triglyceride storage in the adipose tissues, liver, heart, and pancreas [6]. Alterations in the gut microbiome also lead to inflammation, which is a condition that is associated with weight gain and insulin resistance, thus causing impaired glucose tolerance [7]. Behavior alterations (change in appetite and energy expenditure) also occur because of an imbalance in gut microbiota [8].

Currently, antibiotic use has become widespread and is associated with alterations in the intestinal microbiome. Several studies have evaluated the long-term effects of antibiotics. Research in animal models showed that antibiotic treatment could significantly increase obesity risk, and a study found that antibiotic treatment in mice before weaning induced obesity in adulthood [9]. Nevertheless, such evidence in humans remains controversial, as some studies did not report any significant association between antibiotic treatment and obesity risk in childhood [10,11]. In contrast, other studies, including cross-sectional and cohort studies, found that antibiotic treatment can significantly increase obesity risk in childhood [12,13,14]. In adults, a few observational studies have shown an association between antibiotic use and obesity. A French study showed that weight gain occurred after a six-week intravenous treatment with vancomycin and gentamycin for infectious endocarditis, which was significant compared to the matched controls [15].

Although some studies have found an association between antibiotic use and obesity, no firm conclusions have been drawn yet. Therefore, in this study, we analyzed the association between antibiotic use and obesity and metabolic syndrome (MS) in a Korean adult population using data from a population-based cohort.

## 2. Materials and Methods

### 2.1. National Health Insurance Service–National Sample Cohort (NHIS-NSC)

Data from the subjects included in this study were extracted from the National Health Insurance Service–National Sample Cohort (NHIS-NSC), which is a population-based cohort in the Republic of Korea. The Korean NHIS is a single healthcare insurance system that covers approximately 97% of the total Korean population. To define the NHIS-NSC, 2% of the total eligible population were randomly selected and stratified according to characteristics such as age, sex, and area from 2002 to 2015 (*n* = 1,108,369) [16]. Therefore, this sample cohort can be considered representative of the Korean population.

The NHIS-NSC consists of four datasets, including sociodemographic data, medical claims data consisting of data on diagnosis and medication, data collected through a national health screening, and information on medical institutions. Information on diagnoses is represented by the diagnostic codes of the International Classification of Diseases, 10th revision (ICD-10). The national health screening program occurs every two years for all beneficiaries aged ≥40 years. This screening program includes questionnaires on medical history, laboratory tests, anthropometry, and information related to lifestyle such as smoking and alcohol consumption.

### 2.2. Study Subjects

Subjects who maintained their national health insurance or medical care status in 2015 were included (*n* = 273,853). Subjects aged <20 years and those with missing data were excluded. Finally, 266,447 individuals were included in this study (Figure 1). Subjects were divided into four groups according to their obesity status using body mass index (BMI) and waist circumference (WC) criteria of MS (BMI ≥ 25 kg/m^2^_,_ WC ≥ 90/85 cm) [17], namely: obesity 1 group (*n* = 51,880), subjects who met BMI and WC criteria; obesity 2 group (*n* = 103,609), subjects who met BMI or WC criteria; obesity 3 group (*n* = 94,890), subjects who met only BMI criteria; and obesity 4 group (*n* = 60,599), subjects who met only WC criteria. We retrospectively reviewed our cohort during the previous 10 years to evaluate the effect of systemic antibiotic treatment on obesity and MS.

### 2.3. Data and Measurements

Using the datasets of the NHIS-NSC, the following variables were analyzed for the 266,447 subjects: demographic data (age and sex), lifestyle data such as smoking (current smoker, never/ex-smoker), alcohol consumption (heavy drinkers were defined as people who consumed 30 g of alcohol per day) [18], and physical activity (regular exercise was defined as performance of intense exercise for at least 20 min once weekly) [16,19]. Subsequently, laboratory parameters were obtained from the data of the National Health Screening Program: fasting glucose (FG), high-density lipoprotein cholesterol (HDL-C), and triglycerides (TG). In our analysis, we also included information on BMI, WC, and blood pressure (BP). Data on DM, HTN, and dyslipidemia were collected using ICD-10 diagnostic codes and current treatments. The criteria for MS were defined as follows: BMI ≥ 25 kg/m^2^, WC ≥ 90/85 cm, BP ≥ 130/85 mmHg, FG ≥ 100 mg/dL, TG ≥ 150 mg/dL, and HDL < 40/< 50 mg/dL [17,20].

Regarding systemic antibiotic prescriptions in the previous 10 years from 2015, we analyzed the drug classification number. The drug classification number is the number assigned to the items categorized by major drug efficacy and detailed efficacy in Korea. In our analysis, we included antibiotics (those for Gram-positive infections, Gram-negative infections, and other infections) that are used for diseases with the following codes: 610, 611, 612, 613, 614, 615, 616, 618, and 619 (Appendix A Appendix A). Antibiotic users were defined as cases in which the drug classification number occurred at least once. The total duration (days) of antibiotic treatment was defined as the sum of the days of antibiotic administration. Based on the total duration of antibiotic treatment, we divided our sample into four groups: non-users (*n* = 5937), 1st (*n* = 87,886), 2nd (*n* = 86,012), and 3rd tertile users (*n* = 86,612).

### 2.4. Statistical Analysis

To describe the general characteristics of subjects according to obesity and the duration of systemic antibiotic treatment (non-users, 1st, 2nd, and 3rd tertile), we applied a chi-squared test for categorical variables and a Student’s t-test or ANOVA for continuous variables. Finally, we performed several logistic models to analyze the risk of obesity and MS according to the duration of systemic antibiotic treatment. The effect of systemic antibiotics on obesity/components of MS was corrected for several variables, including age, sex, drinking, smoking, physical activity, and presence of comorbidities such as DM, HTN, and dyslipidemia.

## 3. Results

### 3.1. Clinical Characteristics of the Subjects According to Obesity Status

The results are presented in Table 1. Subjects classified as obesity according to BMI or WC were older than those in the non-obesity group (50.9 ± 13.8 vs. 48.4 ± 14.1%, *p* < 0.0001) and had a higher proportion of men (47.35% vs. 62.11%, *p* < 0.0001). Subjects in the obesity group had more comorbidities than those in the non-obesity group (DM, 15.5% vs. 7.55%; HTN, 40.55% vs. 20.10%; dyslipidemia, 34.34% vs. 19.93%; *p* < 0.0001 for all). Additionally, participants in the obesity group were more frequently classified as heavy drinkers (52.25% vs. 48.84%, *p* < 0.0001), current smokers (23.83% vs. 20.01, *p* < 0.0001), and had more regular physical activity (50.22% vs. 49.39%, *p* < 0.0001). More subjects in the obesity group met the criteria for MS than those in the non-obesity group (BP, 61.21% vs. 35.02%; FG, 47.28% vs. 29.61; TG, 54.47% vs. 28.33%; HDL, 42.23% vs. 25.21%; *p* < 0.0001 for all). Finally, the duration of systemic antibiotic treatment for the previous 10 years in the obesity group was longer than that in the non-obesity group (72.3 ± 86.0 vs. 69.4 ± 82.5 days, *p* < 0.0001).

### 3.2. Clinical Characteristics of the Subjects Classified According to Total Duration of Systemic Antibiotic Treatment for the Previous 10 Years

The clinical characteristics of the subjects classified according to the total duration of systemic antibiotic treatment are presented in Table 2. Subjects in 3rd tertile group were the oldest (3rd tertile: 52.2 ± 14.2 years; 2nd tertile: 48.9 ± 14.0 years; 1st tertile: 47.1 ± 13.5 years; non-users: 47.8 ± 12.8 years; *p* < 0.0001) and had the most comorbidities compared to other groups (DM: 12.52%, HTN: 32.86%, dyslipidemia: 30.14%). Subjects who never used antibiotics in the previous 10 years were more frequently classified as current smokers (non-users, 1st, 2nd, and 3rd tertiles: 33.8%, 27.9%, 20.6%, and 15%, respectively; *p* < 0.0001), heavy drinkers (non-users, 1st, 2nd, and 3rd tertiles: 59.1%, 57.1%, 51.1%, and 41.7%, respectively; *p* < 0.0001), and performed more regular physical activity (non-users, 1st, 2nd, and 3rd tertiles: 54.1%, 51.6%, 49.7%, and 47.5%, respectively; *p* < 0.0001). Subjects who used antibiotics for longer periods had higher levels of triglycerides, lower HDL-C levels, larger WC, and met the MS criteria (WC: 24.6% vs. 21.5%; HDL: 38.4% vs. 22.8%; TG: 41.2% vs. 37.7%, in the 3rd tertile group vs. non-users group, respectively; *p* < 0.0001 for all). Conversely, BMI, BP, and FG were higher in the non-user group than in the antibiotic users.

### 3.3. Risk of Obesity and Components of MS According to Total Duration of Systemic Antibiotic Treatment for the Previous 10 Years

The results of the risk of obesity according to the total duration of systemic antibiotic treatment are presented in Table 3. By analyzing the univariate logistic models without the covariates, we found that subjects who took antibiotics for a longer duration (3rd tertile group) had a higher obesity risk than non-users in the obesity 1 and obesity 4 groups (obesity 1: OR (odds ratio) = 1.10, 95% CI (confidence interval) = 1.03–1.18; obesity 4: OR = 1.19, 95% CI = 1.12–1.27). When the effect of antibiotic prescription on obesity risk was corrected for some covariates, such as age and sex, more significant associations were found. Subjects who used systemic antibiotics for longer durations (3rd tertile group) had a higher obesity risk than non-users (obesity 1: OR = 1.26, 95% CI = 1.18–1.35; obesity 2: OR = 1.26, 95% CI= 1.19–1.33; obesity 3: OR = 1.22, 95% CI = 1.15–1.29; obesity 4: OR = 1.28, 95% CI = 1.20–1.37). In particular, when we corrected for the effect of systemic antibiotics on obesity risk for all covariates (age, sex, presence of DM, HTN, dyslipidemia, drinking, smoking, and physical exercise), we found that subjects who took antibiotics for more days had an obesity risk approximately 20% higher than that in non-users (obesity 1, 3rd tertile vs. non-users: OR = 1.20, 95% CI = 1.12–1.38; obesity 2, 3rd tertile vs. non-users: OR = 1.20, 95% CI = 1.13–1.26). Finally, when we analyzed trends in obesity risk (obesity 1–4) according to total duration (days) of antibiotic treatment (non-users, 1st tertile, 2nd tertile, 3rd tertile), we found an increasing linear trend showing that a longer duration of systemic antibiotic treatment therapy can increase the risk of obesity (Figure 2).

Regarding the components of MS, we found that subjects classified in the 3rd tertile group had a higher risk of having MS due to altered levels of TG and HDL-C than non-users (TG criteria for MS, 3rd tertile vs. non-users: OR = 1.21, 95% CI = 1.13–1.29; HDL-C criteria for MS, 3rd tertile vs. non-users: OR = 1.50, 95% CI = 1.39–1.61). Additionally, when we analyzed subjects who took antibiotics for longer durations, we found that they had an increased risk of having MS according to BMI, WC, TG, and HDL criteria (3rd tertile vs. non-users: OR = 1.59, 95% CI = 1.44–1.76).

Finally, when we selected subjects who took antibiotics targeting Gram-negative and Gram-positive bacteria, we found that people who took both types of antibiotics for longer durations had a higher obesity risk (OR > 1 for obesity 1–4 in Gram-positive and Gram-negative antibiotic groups). This association was stronger in subjects who used antibiotics against Gram-negative bacteria (Appendix A Appendix A).

## 4. Discussion

In this study, the duration of systemic antibiotic treatment in the previous 10 years was longer in the obesity group than in the non-obesity group. As a result of dividing the groups according to total duration of systemic antibiotic treatment, the group who used more antibiotics was older and the proportion of women was higher. As one ages, immune function weakens and the number of comorbidities increase, making one more susceptible to infection [21,22]. This can lead to an increased use of antibiotics. In fact, DM, HTN, and hyperlipidemia were more frequent in the group that used antibiotics for the longest duration in our study. To date, there has been no evidence of gender differences influencing the use of antibiotics. This difference may have occurred because our study included only systemic antibiotics and excluded oral antibiotics. Further analysis is needed, with the inclusion of oral antibiotics in the future. The longer the duration of systemic antibiotic treatment in the previous 10 years, the less alcohol consumption, smoking, and regular exercise performed. Since systemic antibiotic treatment is related to hospitalization, lifestyle practices may change due to poor body conditions. Changes in TG, HDL-C, and WC were most pronounced in the group with the longest duration of systemic antibiotic treatment. In fact, since the prevalence of HTN and DM was high in the group with a long duration of systemic antibiotic treatment, changes in BP and fasting glucose may have been affected by medications. In contrast, dyslipidemia medications tend to be widely used to control low-density lipoprotein cholesterol; therefore, a difference in TG and HDL-C may have been observed.

The group who used antibiotics for a longer period had a higher risk of obesity, higher TG, and lower HDL-C levels than the non-users in this study. The mechanisms related to antibiotic effects on weight gain are not clear; however, some hypotheses have been proposed: antibiotics can increase the ability of gut bacteria to extract energy from indigestible polysaccharides [23] and decrease the number of bacteria that are metabolically protective against obesity [24]. Antibiotics can also alter hepatic lipogenesis and metabolism, as well as decrease intestinal defenses and immunity [25].

Systemic antibiotic treatment is more likely to be associated with hospitalization and severe disease compared to oral antibiotic treatment. Patients with serious illness are more likely to be cachexic [26], which is contrary to our findings. This is because our study was designed as a retrospective study and reviewed 10 years of data. Thus, it is highly likely that patients with serious illnesses have already died. Therefore, many patients with mild disease severities may have been included, and they are likely to have lived in the same pattern before and after their disease was cured.

In this study, we found a stronger association with obesity when antibiotics targeting Gram-negative bacteria were used. Gram-negative bacteria have an outer membrane that protects bacteria against toxic molecules such as antibiotics [27], which is composed of glycolipid lipopolysaccharide (LPS). High levels of LPS in the bloodstream have been associated with several diseases, such as obesity, diabetes, and cardiovascular diseases [28]. Although LPS is attached to the bacteria’s outer membrane, it can be released when antibiotics attack the bacteria [29]. This could explain why the use of antibiotics targeting Gram-negative bacteria is strongly associated with obesity.

This study has some limitations. The data were obtained based on insurance claims, which can be flawed because information from insurance databases are often limited and inaccurate. Our data were analyzed retrospectively, with such studies having more confounding effects than prospective ones. Additionally, this is an observational study; therefore, it was impossible to understand the mechanisms that could explain the effects of antibiotics on obesity. Since our study only considered the use of systemic antibiotics, the results of this study cannot be generalized to other types of antibiotics. Further large-scale studies including oral antibiotics are needed in the future.

Nevertheless, this is the first study performed on a Korean adult population to evaluate the association between antibiotics and obesity. Because the effect of antibiotics on obesity risk is still controversial, our results could provide useful information and encourage further research on the subject. This was a large-scale study, and the NHIS sample cohort used in this study was carefully designed and representative of the general population. Additionally, because of the positive association between the use of antibiotics and obesity risk, we emphasize the importance of balancing the risks and benefits of antibiotic therapy.

## 5. Conclusions

This study suggests that antibiotic treatment can affect obesity status and metabolic syndrome components, with this effect being even more prominent with the use of antibiotics targeting Gram-negative bacteria.

## Figures and Tables

**Figure 1 jcm-10-02601-f001:**
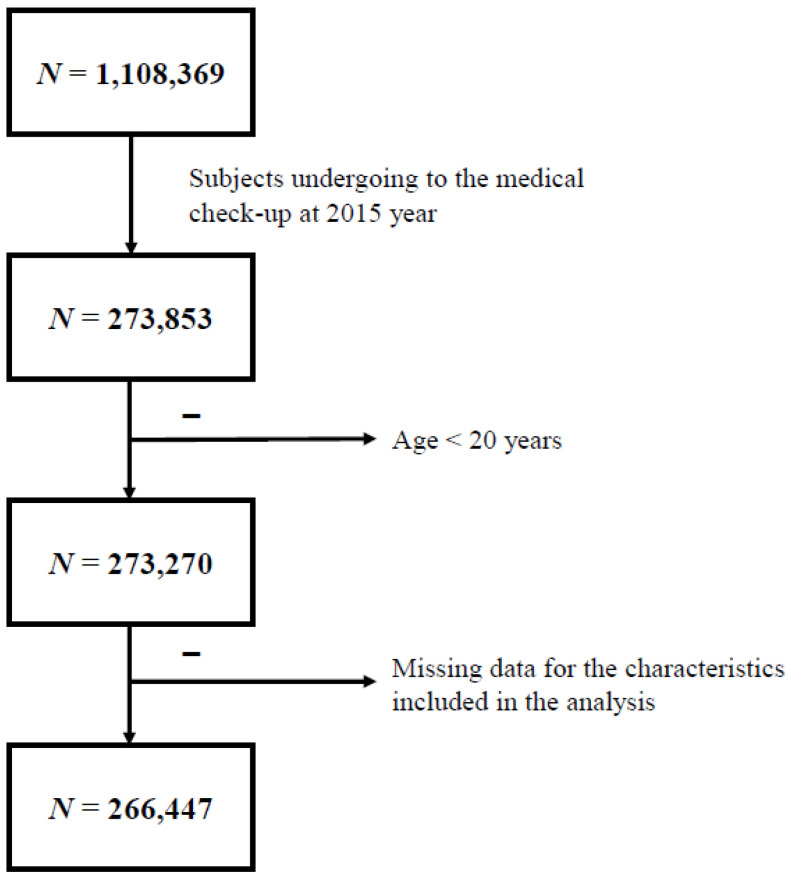
Study progression.

**Figure 2 jcm-10-02601-f002:**
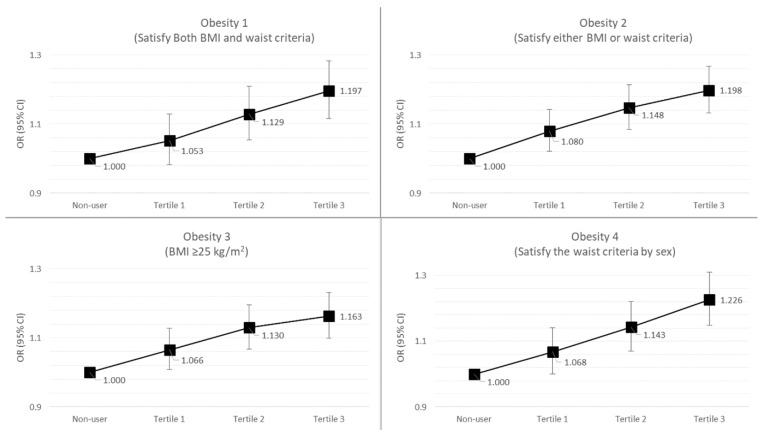
Trends in the risk of obesity according to the duration of systemic antibiotic treatment. BMI, body mass index; OR, odds ratio; CI, confidence interval.

**Table 1 jcm-10-02601-t001:** Characteristics of the study subjects.

Variables	Non-Obesity (*n* = 162,838)	Obesity * (*n* = 103,609)	*p*
Age (years)	48.4 ± 14.1	50.9 ± 13.8	<0.0001
Sex, *n* (%)			<0.0001
Male	77,109 (47.4)	64,351 (62.1)	
Female	85,729 (52.6)	39,258 (37.9)	
Diabetes mellitus, *n* (%)	12,290 (7.55)	16,057 (15.5)	<0.0001
Hypertension, *n* (%)	32,732 (20.1)	42,012 (40.6)	<0.0001
Dyslipidemia, *n* (%)	32,457 (19.9)	35,577 (34.3)	<0.0001
Heavy drinker, *n* (%)	79,523 (48.8)	54,140 (52.3)	<0.0001
Current smoker, *n* (%)	32,592 (20.0)	24,686 (23.8)	<0.0001
Regular exercise, *n* (%)	80,420 (49.4)	52,028 (50.2)	<0.0001
Subject who satisfy MS criteria			
BP ≥ 130/85 mmHg	57,024 (35.0)	63,417 (61.2)	<0.0001
FG ≥ 100 mg/dL	48,220 (29.6)	48,990 (47.3)	<0.0001
TG ≥ 150 mg/dL	46,138 (28.3)	56,437 (54.5)	<0.0001
HDL-C < 40/50 mg/dL	41,052 (25.2)	43,752 (42.2)	<0.0001
Systemic antibiotic treatment duration (days) in the previous 10 years	69.4 ± 82.5	72.3 ± 86.0	<0.0001

* Obesity is defined as meeting the body mass index or waist circumference criteria. Mean ± SD, chi-squared test, and Student’s *t*-test; BP, blood pressure; MS, metabolic syndrome; FG, fasting glucose; TG, triglycerides; HDL-C, high-density lipoprotein cholesterol.

**Table 2 jcm-10-02601-t002:** Characteristics of the subjects classified according to the total duration of 10 years (all antibiotics).

Variables	Non-Users (*n* = 5937)	1st Tertile (*n* = 87,886)	2nd Tertile (*n* = 86,012)	3rd Tertile (*n* = 86,612)	*p*
Age (years)	47.8 ± 12.8	47.1 ± 13.5	48.9 ± 14.0	52.2 ± 14.2	<0.0001
Sex, *n* (%)					<0.0001
Male	4632 (78.0%)	56,745 (64.6%)	43,982 (51.1%)	36,101 (41.7%)	
Female	1305 (22.0 %)	31,141 (35.4%)	42,030 (48.9%)	50,511 (58.3%)	
Diabetes mellitus, *n* (%)	597 (10.1%)	8214 (9.4%)	8693 (10.1%)	10,843 (12.5%)	<0.0001
Hypertension, *n* (%)	1599 (26.9%)	21,619 (24.6%)	23,067 (26.8%)	28,459 (32.9%)	<0.0001
Dyslipidemia, *n* (%)	1230 (20.72%)	19,282 (21.9%)	21,415 (24.9%)	26,107 (30.1%)	<0.0001
Heavy drinker, *n* (%)	3510 (59.1%)	50,151 (57.1%)	43,913 (51.1%)	36,089 (41.7%)	<0.0001
Current smoker, *n* (%)	2008 (33.8%)	24,557 (27.9%)	17,718 (20.6%)	12,995 (15.0%)	<0.0001
Regular exercise, *n* (%)	3211 (54.1%)	45,332 (51.6%)	42,731 (49.7%)	41,174 (47.5%)	<0.0001
Subject who satisfy criteria of MS					
BMI ≥ 25 kg/m^2^	2146 (36.2%)	31,234 (35.5%)	30,437 (35.4%)	31,073 (35.9%)	0.1401
WC ≥ 90/85 cm	1276 (21.5%)	18,825 (21.4%)	19,209 (22.3%)	21,289 (24.6%)	<0.0001
BP ≥ 130/85 mmHg	2919 (49.2%)	38,864 (44.2%)	37,516 (43.6%)	41,142 (47.5%)	<0.0001
FG ≥ 100 mg/dL	2441 (41.1%)	31,920 (36.3%)	30,568 (35.5%)	32,281 (37.3%)	<0.0001
TG ≥ 150 mg/dL	2238 (37.7%)	32,532 (37.0%)	32,106 (37.3%)	35,699 (41.2%)	<0.0001
HDL-C < 40/50 mg/dL	1353 (22.8%)	23,305 (26.5%)	26,896 (31.3%)	33,250 (38.4%)	<0.0001

*n* (%) or mean ± SD, chi-squared test, and ANOVA. BMI, body mass index; WC, waist circumference; BP, blood pressure; MS, metabolic syndrome; FG, fasting glucose; TG, triglycerides; HDL-C, high-density lipoprotein cholesterol.

**Table 3 jcm-10-02601-t003:** Results of logistic models for the risk of obesity and some components of MS according to antibiotic treatment duration (days).

	Antibiotics Prescription Days for 10 Years			
	Non-Users (*n* = 5937)	1st Tertile (*n* = 87,886)	2nd Tertile (*n* = 86,012)	3rd Tertile (*n* = 86,612)
Obesity 1 *, OR (95%CI)				
Event	1129	16,422	16,515	17,814
Model 1 **	1.00 (ref)	0.98 (0.92; 1.05)	1.01 (0.95; 1.08)	1.10 (1.03; 1.18)
Model 2 **	1.00 (ref)	1.06 (0.99; 1.13)	1.15 (1.07; 1.23)	1.26 (1.18; 1.35)
Model 3 **	1.00 (ref)	1.05 (0.98; 1.13)	1.13 (1.05; 1.21)	1.19 (1.11; 1.28)
Model 4 **	1.00 (ref)	1.05 (0.98; 1.13)	1.13 (1.05,1.21)	1.20 (1.12,1.38)
Obesity 2 *, OR (95%CI)				
Event	2293	33,637	33,131	34,548
Model 1 **	1.00 (ref)	0.99 (0.93; 1.04)	0.99 (0.94; 1.05)	1.06 (0.99; 1.11)
Model 2 **	1.00 (ref)	1.08(1.03; 1.14)	1.17(1.10; 1.23)	1.26 (1.19; 1.33)
Model 3 **	1.00 (ref)	1.08 (1.02; 1.14)	1.15 (1.09; 1.22)	1.19 (1.13; 1.27)
Model 4 **	1.00 (ref)	1.08 (1.02; 1.14)	1.15 (1.09; 1.21)	1.20 (1.13; 1.26)
Obesity 3 *, OR (95%CI)				
Event	2146	31,234	30,437	31,073
Model 1 **	1.00 (ref)	0.97 (0.92; 1.03)	0.97 (0.92; 1.02)	0.99 (0.94; 1.04)
Model 2 **	1.00 (ref)	1.07 (1.01; 1.13)	1.15 (1.09; 1.21)	1.22 (1.15; 1.29)
Model 3 **	1.00 (ref)	1.07 (1.01; 1.13)	1.13 (1.07; 1.19)	1.16 (1.10; 1.23)
Model 4 **	1.00 (ref)	1.07 (1.01; 1.13)	1.13 (1.07; 1.19)	1.16 (1.09; 1.23)
Obesity 4 *, OR (95%CI)				
Event	1276	18,825	19,209	21,289
Model 1 **	1.00 (ref)	0.99 (0.93; 1.06)	1.05 (0.99; 1.12)	1.19 (1.12; 1.27)
Model 2 **	1.00 (ref)	1.07 (1.01; 1.14)	1.16 (1.086,1.236)	1.28 (1.20; 1.37)
Model 3 **	1.00 (ref)	1.07 (0.99; 1.14)	1.14 (1.07; 1.22)	1.22 (1.14; 1.30)
Model 4 **	1.00 (ref)	1.07 (1.00; 1.14)	1.14 (1.07; 1.22)	1.23 (1.15; 1.31)
TG ≥ 150 mg/dL, OR (95%CI)				
Event	2238	32,532	32,106	35,699
Model 1 **	1.00 (ref.)	0.97 (0.92; 1.03)	0.98 (0.93; 1.04)	1.16 (1.09; 1.22)
Model 2 **	1.00 (ref.)	1.09 (1.04; 1.16)	1.15 (1.08; 1.21)	1.29 (1.23; 1.37)
Model 3 **	1.00 (ref.)	1.08 (1.01; 1.15)	1.09 (1.02; 1.16)	1.17 (1.09; 1.25)
Model 4 **	1.00 (ref.)	1.09 (1.02; 1.16)	1.11 (1.04; 1.19)	1.21 (1.13; 1.29)
HDL < 40/<50 mg/dL, OR (95%CI)				
Event	1353	23,305	26,896	33,250
Model 1 **	1.00 (ref.)	1.22 (1.15; 1.30)	1.54 (1.44; 1.64)	2.11 (1.98; 2.25)
Model 2 **	1.00 (ref.)	1.22 (1.14; 1.31)	1.38 (1.29; 1.47)	1.60 (1.49; 1.71)
Model 3 **	1.00 (ref.)	1.21 (1.12; 1.30)	1.34 (1.24; 1.44)	1.48 (1.39; 1.60)
Model 4 **	1.00 (ref.)	1.23 (1.14; 1.32)	1.36 (1.27; 1.46)	1.50 (1.39; 1.61)
Satisfy BMI, WC, TG, HDL-C criteria of MS, OR (95%CI)				
Event	542	9552	11,000	14,304
Model 1 **	1.00 (ref.)	1.21 (1.11; 1.32)	1.46 (1.33; 1.59)	1.97 (1.79; 2.16)
Model 2 **	1.00 (ref.)	1.28 (1.17; 1.40)	1.47 (1.34; 1.61)	1.79 (1.63; 1.96)
Model 3 **	1.00 (ref.)	1.25 (1.13; 1.38)	1.39 (1.26; 1.54)	1.59 (1.44; 1.76)
Model 4 **	1.00 (ref.)	1.26 (1.14; 1.39)	1.41 (1.27; 1.55)	1.59 (1.44; 1.76)

* Obesity 1: subjects who met both BMI and WC criteria; * obesity 2: subjects who met BMI or WC criterion; * obesity 3: subjects who met only BMI criterion; * obesity 4: subjects who met only WC criterion. ** Model 1: non-adjusted; ** Model 2: adjusted for age and sex; ** Model 3: adjusted for factors shown in Model 2 and comorbidity such as DM, HTN, and dyslipidemia; ** Model 4: adjusted for factors shown in Model 3 and lifestyle factors such as drinking, smoking, and exercise. OR, odds ratio; CI, confidence interval.

## Data Availability

Data available on request from corresponding authors.

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
