# Peer review of "Systemic Antibiotics and Obesity: Analyses from a Population-Based Cohort"

_jcm, 2021, doi:10.3390/jcm10122601_

Round 1

Reviewer 1 Report

The introduction to the work is very clear, the authors cite good-quality scientific articles, characterizing the problem and introducing the reader to the subject of the article. The work methodology does not require significant modifications. The criteria for enrolling patients in the study and the classification to specific groups are described in a clear and understandable way. Only there is no mention of the use of probiotic therapy by patients (combined or not) with antibiotic therapy - this information could significantly change the way of drawing conclusions in this work.

Please also fill in the number of people in each group in methodology section. I am also wondering about the intensity of physical activity described at work - as one 20-minute unit? This is a very low level of effort per week - why did the authors choose this criterion?

The limitations of work described by the authors are noteworthy. Presenting the methodology and results of work in a critical manner allows for better planning and implementation of further scientific research in the future. The thesis conclusions are correct and do not require editing.

After minor changes to the article by the authors, the work can be published.

Author Response

1.The introduction to the work is very clear, the authors cite good-quality scientific articles, characterizing the problem and introducing the reader to the subject of the article. The work methodology does not require significant modifications. The criteria for enrolling patients in the study and the classification to specific groups are described in a clear and understandable way. Only there is no mention of the use of probiotic therapy by patients (combined or not) with antibiotic therapy - this information could significantly change the way of drawing conclusions in this work.

→ Thank you for your comments. I totally agree with your opinion that probiotic therapy will have an effect. The data we used is Korean medical insurance claim data. Unfortunately, in Korea, probiotics are often prescribed without insurance, and in many cases, they are prescribed arbitrarily at pharmacies, so it is difficult to accurately identify them. I think your opinion is very important, and I think that in future research, a delicate design that includes a questionnaire on whether or not to take probiotics is needed.

2.Please also fill in the number of people in each group in methodology section. I am also wondering about the intensity of physical activity described at work - as one 20-minute unit? This is a very low level of effort per week - why did the authors choose this criterion?

→ As you said, I filled in the number of people in each group in methodology section. That regular exercise was defined as performance of intense exercise for at least 20 minutes once weekly may be very low level of effort per week according to your opinion. When the National Health Insurance Service – National Sample Cohort (NHIS-NSC) that we used conducted a questionnaire about exercise, it was investigated how many times a week did you exercise intensively more than 20 minutes. This is also a limitation of the NHIS-NSC we used.

Reference

Lee, J.; Lee, J.S.; Park, S.H.; Shin, S.A.; Kim, K. Cohort Profile: The National Health Insurance Service-National Sample Cohort (NHIS-NSC), South Korea. Int J Epidemiol 2017, 46, e15, doi:10.1093/ije/dyv319.

3.The limitations of work described by the authors are noteworthy. Presenting the methodology and results of work in a critical manner allows for better planning and implementation of further scientific research in the future. The thesis conclusions are correct and do not require editing.

→ Thank you very much for your kind comments.

Reviewer 2 Report

In this retrospective study, the association between antibiotic use and obesity and metabolic syndrome was analyzed in 266,447 Korean adult individuals in the year 2015.

The Researchers conclude that the risk of Obesity is higher in subjects tooking systemic antibiotics for longer periods in the previous 10 years and it results more prominent when they took antibiotics targeting gram-negative bacteria. Moreover these subjects had an increased risk of developing Metabolic Syndrome.

From the methodological point of view, the study is adequately appropriate and suitable to scientific rigour and ethical principles. The study design process is valid, both inclusion and exclusion criteria are compliant, as well as data collection; the statistical analysis seems valid; purposes and scientific contents are supported by valid References; editorial structure, figure and tables are good. The results are clear and the “Discussion” section is well structured and interesting.

If these data will be confirmed in other worldwide studies including also oral antibiotics, they would improve pathway but, overall, prevention, of weight gain and its metabolic complications. In this way, the effort of the Authors will be remarkable and I think this research will be a useful tool for other large scale studies and it will stimulate other Researchers.  

I believe that the manuscript is overall valid and deserves publication in the "Journal of Clinical Medicine".

Author Response

Thank you for your kind comments. According to your opinion, a more delicately designed follow-up study including oral antibiotics is needed, which will show great meaningful results.